# Analysis and Prediction of Pedestrians' Violation Behavior at the Intersection Based on a Markov Chain

**Chengyuan Mao \*, Lewen Bao, Shengde Yang, Wenjiao Xu and Qin Wang**

Road and Traffic Engineering Institute, Zhejiang Normal University, Jinhua 321004, China; baolwsjtu@163.com (L.B.); yangsd@zjnu.edu.cn (S.Y.); xuwj@zjnu.edu.cn (W.X.); 18816238356@163.com (Q.W.)
\* Correspondence: maocy@zjnu.cn; Tel.: +86-15-067-997774

**Abstract:** Pedestrian violations pose a danger to themselves and other road users. Most previous studies predict pedestrian violation behaviors based only on pedestrians' demographic characteristics. In practice, in addition to demographic characteristics, other factors may also impact pedestrian violation behaviors. Therefore, this study aims to predict pedestrian crossing violations based on pedestrian attributes, traffic conditions, road geometry, and environmental conditions. Data on the pedestrian crossing, both in compliance and in violation, were collected from 10 signalized intersections in the city of Jinhua, China. We propose an illegal pedestrian crossing behavior prediction approach that consists of a logistic regression model and a Markov Chain model. The former calculates the likelihood that the first pedestrian who decides to cross the intersection illegally within each signal cycle, while the latter computes the probability that the subsequent pedestrians who decides to follow the violation. The proposed approach was validated using data gathered from an additional signalized intersection in Jinhua city. The results show that the proposed approach has a robust ability in pedestrian violation behavior prediction. The findings can provide theoretical references for pedestrian signal timing, crossing facility optimization, and warning system design.

**Keywords:** signalized intersection; pedestrian violation; Markov chain; logistic regression

## 1. Introduction

As the key node of the urban road network [1], road intersections undertake the important task of separating the participants of various traffic modes. At present, mixed traffic is the main feature of urban traffic in China [2]. In order to reduce the mutual interference between pedestrians and motor vehicles, crosswalks and signal lights are usually set at intersections [3]. Relevant studies show that the proportion of traffic accidents caused by running red lights at crosswalks is more than 40%, and the annual death toll is up to thousands [4]. The increasingly serious problem of pedestrian traffic not only poses a great threat to the safety of people's lives and property, but also seriously disrupts the social order.

Affected by the herd mentality, the violation behavior of pedestrians is basically completed in the form of groups. Generally speaking, the first pedestrian crossing the street will comprehensively judge the situation of the intersection and implement the corresponding behavior decision, while the subsequent pedestrian crossing the street will show a judgment and understanding in conformity with public opinion or the behavior of the majority under the influence of herd mentality. Studies have shown [5] that the herd psychology can prompt pedestrians who cross the street illegally to form a temporary group and influence the decision-making of each individual in the group to make the same choice.

By combing through the related literature at home and abroad, a large number of previous analysis and research have been conducted on pedestrian crossing violation prediction. Mihiran et al. [6] proposed a combination of a Markov model and a Bayesian network

for a real-time monitoring method to estimate the probability of pedestrian group violations. Saeideh et al. [7] applied a region growing technique to construct a Hidden Markov Model (HMM) to simulate the time-varying trajectory of moving objects. Javan D. et al. [8] analyzed polydisperse systems by a Markov chain algorithm, but ignored the influence of interaction between groups. Considering the great uncertainty in pedestrian crossing decisions, this paper combines a logistic model and a Markov chain to build a pedestrian crossing violation model, which will improve prediction accuracy.

To sum up, this paper focuses on the phenomenon and probability of pedestrians crossing the crosswalk illegally at signalized intersections from the perspective of crowd psychology. A quantitative prediction model of the decision-making behavior acts by the first pedestrian is obtained by logistic regression analysis, and the transition probability matrix between different states is constructed by a Markov chain [9]. The results can be used to analyze and predict the violation probability of pedestrian groups.

## 2. Analysis and Investigation of Pedestrian Crossing Decision Behavior at Signalized Intersection

### 2.1. Research on the Influencing Factors of Pedestrian Violation

According to the law of the Road Traffic Safety Law of the People's Republic of China, pedestrian crossing violations are mainly divided into two types: temporal or spatial, as shown in Figures 1 and 2.

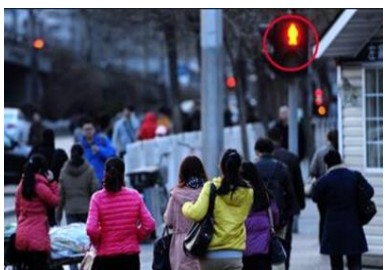

**Figure 1.** Temporal violations by crossing pedestrians.

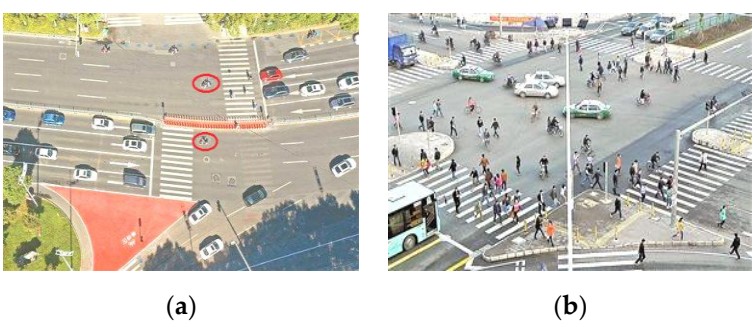

|     (a)     |     (b)     |

**Figure 2.** Spatial violations by crossing pedestrians. (**a**) Pedestrians cross the street twice at the dislocated zebra crosswalks. (**b**) Pedestrians cross the street diagonally at the intersection.

Through a field investigation and review of previous research results, it was found that the decision-making behavior of pedestrians crossing the street is mainly affected by three aspects:

(1)  Road environment

Before modeling and analyzing the traffic flow at intersections, including pedestrian flow and vehicle flow, it is necessary to understand the geometric dimensions of road facilities and signal design. The specific implementation method is to obtain the relevant data information of the collection place through field investigation, including the length and width of the crosswalk and the waiting area, the number of lanes, the signal cycle, and the duration of the red light.

(2)  Crossing facilities

The reasonable layout of pedestrian crossing facilities can improve the comfort of pedestrians using these facilities, and indirectly reduce the probability of pedestrian violation. The pedestrian crossing distances and the type of pedestrian crossing facilities can vary from one signalized intersection to another, which will lead to the occurrence of violations. The crosswalk can be divided into three forms according to the setting of the pedestrian refuge: no pedestrian refuge, pedestrian refuge in the middle of the road, and pedestrian refuge at the right turn ramp, as shown in Figure 3.

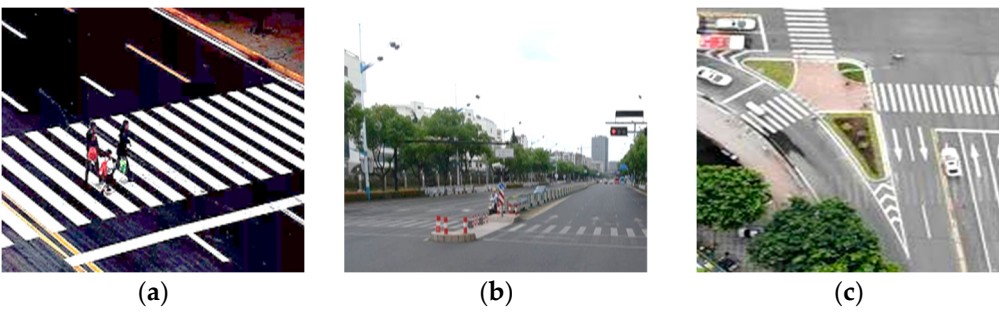

| (a) | (b) | (c) |

**Figure 3.** Three types of crosswalk facilities at level intersections. (**a**) No pedestrian refuge. (**b**) Pedestrian refuge in the middle of the road. (**c**) Pedestrian refuge at the right turn ramp.

(3)  Traffic condition

When pedestrians cross the street at the intersection, they often focus on the acceptable gap to avoid conflict with motor vehicles. In addition, the pedestrian's decision-making behavior will be affected by the waiting time of the red light and whether a countdown device set or not [10].

## 2.2. Index Quantification and Sample Collection

In order to fully cover the factors affecting pedestrian violations, the authors investigated and selected 10 representative intersections in Jinhua City as the research objects, and summarized the easily observed and quantifiable pedestrian violation indicators in the actual survey, as shown in Table 1, and the basic information of each intersection is shown in Table 2. Aimed to improve the efficiency of the survey and increase the sample size, the survey was conducted in the morning and evening rush hours with more pedestrians, 8:00-9:00 a.m. and 17:00-18:00 p.m.

According to the relevant design standards and statistical needs, the indicators are classified as follows:

(1)  Crosswalk width $X_1$. According to the standard "Urban Road Traffic Signs and Markings Setting Specification (GB51038-2015)", the width of the pedestrian crossing is greater than or equal to 3 m, and 1 m is the first level of widening. This paper obtains the actual width of pedestrian crossing through field investigation and measurement, and the classification and coding are as follows: (1) $X_1$ = 3 m or 4 m; (2) $X_1$ = 5 m or 6 m; 3) 6 m < $X_1$.

(2)  Crosswalk type $X_2$. The crosswalks are classified according to the setting form of the intersection, and the codes are as follows: (1) no pedestrian refuge, (2) pedestrian refuge in the middle of the road, and (3) pedestrian refuge at the right turn ramp.

(3)  Intersection environment $X_3$. The surrounding environment when pedestrians are waiting to cross the street has a direct impact on the behavior decision of pedestrians. In this paper, the waiting environment is divided as follows: (1) the inhabiting recreation district, (2) the central business district, and (3) the mixed-use district.

(4)  Traffic congestion $X_4$. According to the service level of signalized intersections, they are classified as (1) smooth (delay d $\leq$ 25 s), (2) delayed (25 s < delay d $\leq$ 60 s), and (3) congested (60 s < delay d).

(5) Average headway $X_5$. When pedestrians make decisions on crossing behavior, they will make a subjective judgment on the headway of motor vehicles. If it is within the safe range, pedestrians with weak traffic awareness may act illegally. In this survey, the headway of motor vehicles is classified as follows: (1) $X_5 \leq 5$ s; (2) 6 s $< X_5 \leq 10$ s; (3) 10 s $< X_5$.

(6) Pedestrian waiting time $X_6$. Whether the setting of a signal lamp duration is reasonable or not will directly affect the decision-making behavior of pedestrians at intersections. Too long a waiting time at a red light will promote the occurrence of pedestrian violations. Classification according to the survey data: (1) 0 s $< X_6 \leq 60$ s; (2) 60 s $< X_6 \leq 120$ s; (3) 120 s $< X_6$.

(7) Red light countdown device $X_7$. The setting of the countdown signal lamp will affect the choice of pedestrian crossing. According to whether the intersection is equipped with red light countdown device, the corresponding setting variables are as follows: (1) with a countdown device and (2) without a countdown device.

Through the survey, 2183 samples of the pedestrian crossing were obtained, including 391 first pedestrian violation samples and 376 following pedestrian violation samples. The statistics of pedestrian violations are shown in Table 3.

**Table 1.** Code and description of pedestrian crossing behavior impact indicators.

| Factors | Influencing Factors | Variable Code and Description |
|---|---|---|
| $X_1$ | Crosswalk width | (1) $X_1 = 3$ m or 4 m; (2) $X_1 = 5$ m or 6 m; (3) 6 m $< X_1$ |
| $X_2$ | Crosswalks type | (1) no pedestrian refuge; (2) pedestrian refuge in the middle of the road; (3) pedestrian refuge at the right turn ramp |
| $X_3$ | Intersection environment | (1) inhabiting recreation district; (2) central business district; (3) mixed-use district |
| $X_4$ | Traffic congestion | (1) smooth; (2) delayed; (3) congested |
| $X_5$ | Average headway | (1) $X_5 \leq 5$ s; 2–6 s $< X_5 \leq 10$ s; 3–10 s $< X_5$ |
| $X_6$ | Pedestrian waiting time | 1–0 s $< X_6 \leq 60$ s; 2–60 s $< X_6 \leq 120$ s; 3–120 s $< X_6$ |
| $X_7$ | Red light countdown device | (1) yes; (2) no |

**Table 2.** Basic traffic environment information of intersections from which samples were collected.

| No. | Investigation Site | $X_1$ | $X_2$ | $X_3$ | $X_4$ | $X_5$ | $X_6$ | $X_7$ |
|---|---|---|---|---|---|---|---|---|
| 1 | Liyu Road-Shuanglong South Street | 1 | 2 | 1 | 3 | 3 | 2 | 2 |
| 2 | Danxi Road-Shuanglong South Street | 1 | 2 | 3 | 2 | 1 | 3 | 1 |
| 3 | Danxi Road-Lanxi Street | 2 | 3 | 2 | 2 | 1 | 3 | 1 |
| 4 | Jiefang West-Road Huixi Street | 2 | 1 | 3 | 3 | 2 | 1 | 2 |
| 5 | Bayi North Street-Jiefang East Road | 3 | 1 | 1 | 1 | 1 | 1 | 1 |
| 6 | Jiefang East Road-Xinhua Street | 3 | 1 | 1 | 3 | 1 | 1 | 2 |
| 7 | North Second Ring Road-Yingbin West Avenue | 2 | 3 | 3 | 2 | 3 | 3 | 2 |
| 8 | Danguang West Road-Yingbin Avenue | 3 | 2 | 2 | 1 | 1 | 3 | 1 |
| 9 | Jiangjun Road-Shengli Street | 2 | 1 | 2 | 1 | 2 | 1 | 1 |
| 10 | Dongshi South Street-Liyu East Road | 1 | 1 | 1 | 3 | 1 | 2 | 2 |

**Table 3.** Statistics of pedestrian violations.

| Factors | Variable Code | Number of Pedestrians | Number of Violations | | Violation Ratio (%) |
|---|---|---|---|---|---|
| | | | First Pedestrians | Following Pedestrians | |
| $X_1$ | 1 | 460 | 75 | 87 | 35.2 |
| | 2 | 986 | 194 | 175 | 37.4 |
| | 3 | 737 | 122 | 114 | 32.0 |
| $X_2$ | 1 | 1455 | 319 | 305 | 42.9 |
| | 2 | 320 | 45 | 42 | 27.2 |
| | 3 | 408 | 27 | 29 | 18.6 |
| $X_3$ | 1 | 665 | 106 | 103 | 31.4 |
| | 2 | 427 | 47 | 46 | 21.8 |
| | 3 | 1091 | 238 | 227 | 42.6 |
| $X_4$ | 1 | 602 | 75 | 73 | 24.6 |
| | 2 | 941 | 155 | 149 | 32.3 |
| | 3 | 640 | 161 | 154 | 49.2 |
| $X_5$ | 1 | 1603 | 241 | 230 | 29.4 |
| | 2 | 350 | 100 | 97 | 56.3 |
| | 3 | 230 | 50 | 49 | 43.0 |
| $X_6$ | 1 | 1393 | 230 | 221 | 32.4 |
| | 2 | 224 | 55 | 53 | 48.2 |
| | 3 | 566 | 106 | 102 | 36.8 |
| $X_7$ | 1 | 924 | 212 | 202 | 44.8 |
| | 2 | 1259 | 179 | 174 | 28.0 |

As shown in the above table, due to the numerous factors influencing pedestrian crossing decision-making behavior, it is necessary to use the correlation between variables to reduce the dimension of the factors in the raw data, so as to facilitate the use of public factors to process the overall raw data. Using statistical analysis software SPSS for factor analysis, combined with the existing literature research methods, three public factors were extracted, which can cover the valid information required for this study [11,12]. Figure 4 shows the impact index system of pedestrian crossing violation behavior.

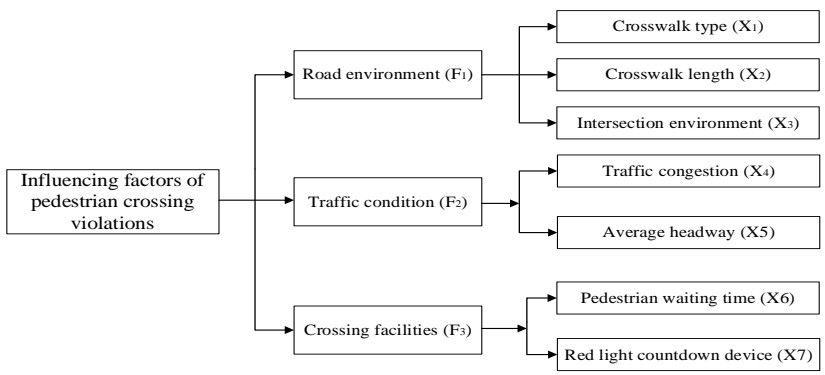

**Figure 4.** The impact index system of pedestrian violation behavior.

The first level indicator is the pedestrian crossing violation probability; the second level indicator, the road environment ($F_1$), reflects the impact of intersection facilities on pedestrian violations, including the crosswalk type ($X_1$), crosswalk width ($X_2$), and

intersection environment ($X_3$); the second level indicator, traffic condition ($F_2$), mainly reflects the impact of the traffic environment at the intersection on pedestrian violations, including traffic congestion ($X_4$) and average headway ($X_5$); the third level indicator, the crossing facility ($F_3$), reflects the impact of signal light-related factors on pedestrian violations, including pedestrian waiting time ($X_6$) and the red light countdown device ($X_7$).

After extracting the influencing factors of pedestrian crossing behavior, the state of each variable is evaluated by the factor analysis model and the factor score coefficient matrix is obtained, as shown in Table 4. The linear combinations of variables can then be used to represent the public factors in the influencing factor system.

**Table 4.** The factor score coefficient matrix.

| Factors | $F_1$ | $F_2$ | $F_3$ |
|---------|-------|-------|-------|
| $X_1$ | 0.260 | −0.062 | 0.013 |
| $X_2$ | 0.441 | −0.411 | 0.011 |
| $X_3$ | 0.329 | 0.306 | −0.042 |
| $X_4$ | 0.039 | 0.529 | 0.154 |
| $X_5$ | 0.121 | 0.034 | −0.038 |
| $X_6$ | 0.026 | −0.061 | 0.383 |
| $X_7$ | −0.094 | −0.042 | 0.274 |

Combined with Table 4, the original variables are represented by the linear combinations of factor variables, and the linear expressions for five public factors are as follows:

$$\begin{cases} F_1 = 0.260X_1 + 0.441X_2 + 0.329X_3 + 0.039X_4 + 0.121X_5 \\ \qquad\quad +0.026X_6 - 0.094X_7 \\ F_2 = -0.062X_1 - 0.411X_2 + 0.306X_3 + 0.529X_4 + 0.034X_5 \\ \qquad\quad -0.061X_6 - 0.042X_7 \\ F_3 = 0.013X_1 + 0.011X_2 - 0.042X_3 + 0.154X_4 - 0.038X_5 \\ \qquad\quad +0.383X_6 + 0.274X_7 \end{cases} \tag{1}$$

## 3. Model Construction

Based on the survey results obtained in Section 2, the prediction model of the first pedestrian decision-making behavior is constructed by using the logistic regression analysis method. In order to further explore the influence of herd mentality on pedestrian decision-making behavior, the transition probability matrix of subsequent pedestrian group violations is calculated by a Markov chain method, which can analyze and predict the crossing decision-making behavior of the whole pedestrian group at the intersection.

### 3.1. Decision Model of Pedestrian Crossing Behavior Based on Logistic Regression

In order to determine whether the expected frequencies are significantly different from the observed frequencies, a goodness-of-fit test of the logistic regression model is required. In this paper, the likelihood ratio test is carried out on the frequency of the occurrence and non-occurrence of model prediction and observation events to determine whether the model is valid, where the likelihood ratio statistics approximately obey the $\chi^2$ distribution. The so-called chi-square model $\chi^2$ can be defined as the gap between the zero hypothesis model [13] and the set model at $-2L\hat{L}$. When the chi-square model $\chi^2$ is large, its significance is small, that is, the difference between the predicted value and the observed value is not obvious, indicating that the model can fit the data well.

$$\chi^2 = \sum \frac{(f_o - f_e)^2}{f_e} \tag{2}$$

where $f_o$ is the actual observation frequency; $f_e$ is the expected frequency.

For two-dimensional interaction tables or univariate logistic regression models, $\chi^2 = 0.05$ is usually used as the significant level for screening candidate variables; that is, when the $\chi^2$ significance value is less than 0.05, the independent variables are valuable for predicting the results [14].

In Section 2, three public factors were extracted by principal component analysis, and their variance contribution rate was as high as 80.147% through computer analysis, which can replace most information of the original seven variables. Before substituting them into the logistic model, this paper also conducts factor screening.

According to Table 5, three independent variables of the road environment, crossing facilities, and traffic conditions all have a freedom degree of 1, and their significance is all less than 0.05, which indicates that they have a significant impact on the decision-making of crossing behavior and can be brought into the model.

**Table 5.** Logistic model likelihood ratio test.

| Independent Variable | Chi-Square | Sig. |
|:---:|:---:|:---:|
| $F_1$ | 11.567 | 0.001 |
| $F_2$ | 13.057 | 0.003 |
| $F_3$ | 60.102 | 0.000 |

When pedestrians cross the crosswalk with signalized control, there are only two kinds of decision-making behaviors: crossing in violation or crossing in compliance, which can be considered as binary variables. Therefore, this paper uses the binary logistic model to fit the data, taking whether or not pedestrians cross in violation as the dependent variable of the model, and various factor indicators as the independent variables. The model defines that the dependent variable code of pedestrian crossing is $y_1 = 1$, and the selection probability is $P_1 = P$; if there is a pedestrian crossing violation, the dependent variable code is $y_1 = 0$, and the selection probability is $P_0 = 1 - P_1$.

The binomial logistic regression model with K independent variables is as follows:

$$p(y_i|x_i) = p_1 = \frac{1}{e^{-(\alpha+\sum_{k=1}^{k}\beta_k x_k)}} \frac{e^{(\alpha+\sum_{k=1}^{k}\beta_k x_k)}}{1+e^{-(\alpha+\sum_{k=1}^{k}\beta_k x_k)}} \tag{3}$$

Therefore, the occurrence ratio of a pedestrian crossing in compliance, that is, the probability of event occurrence and the probability of event non-occurrence, is

$$odds = \frac{p_1}{1-p_1} = e^{(\alpha+\sum_{k=1}^{k}\beta_k x_k)} \tag{4}$$

Take the logarithm on both sides of the above formula,

$$\ln(\frac{p_1}{1-p_1}) = \alpha + \sum_{k=1}^{k}\beta_k x_k \tag{5}$$

After screening the independent variables, three indicators that meet the requirements of a significant level are finally selected. According to the requirements of the logistic regression model, the following regression linear equation can be obtained:

$$\ln(\frac{p_1}{1-p_1}) = \alpha + \beta_1 F_1 + \beta_2 F_2 + \beta_2 F_3 \tag{6}$$

$$p_1 = \frac{1}{1+e^{-(\alpha+\beta_1 F_1+\beta_2 F_2+\beta_2 F_3)}} \tag{7}$$

where $\alpha$ is the constant term; $\beta_j$ is the regression coefficient of each independent variable.

The statistical software SPSS was used to calibrate the impact factor system of pedestrian crossing decision behaviors, using variables satisfying the significance level to estimate the model parameters. The final results are shown in Table 6, where the degrees of freedom of the parameters are all 1:

**Table 6.** Parameter estimation and test results.

| Variable | Coefficient B | Standard Error | Wald | Sig. | Exp (B) |
|---|---|---|---|---|---|
| Intercept | 0.296 | 0.411 | 0.564 | 0.462 | 1.325 |
| F1 | 0.534 | 0.224 | 5.696 | 0.017 | 1.705 |
| F2 | −1.598 | 0.367 | 4.524 | 0.009 | 0.587 |
| F3 | 0.429 | 0.282 | 0.802 | 0.319 | 1.506 |

According to the model estimation results, the pedestrian crossing compliance rate is

$$p_1 = \frac{1}{1 + exp^{-0.293 - 0.534F1 - 0.429F2 + 1.598F3}} \tag{8}$$

$$p_1 + p_0 = 1 \tag{9}$$

After estimating the parameters of the logistic model, deviation statistics, person $\chi^2$, and Hosmer-Lemeshow statistics were used to test the effectiveness of the influence degree of the model coefficient response.

From the data in Table 7, when the significance level $\chi^2 = 0.05$, the selected index variables of pedestrian crossing violations can well fit the data, so the independent variable index of the model has a significant explanatory ability for the dependent variable.

**Table 7.** Logistic goodness of fit test results.

| Goodness of Fit Index | Chi-Square | df | Sig. |
|---|---|---|---|
| Deviance | 157.234 | 82 | 0.543 |
| Person$\chi^2$ | 136.327 | 82 | 0.543 |
| Hosmer-Lemeshow | 26.379 | 14 | 0.248 |

### 3.2. Probability Model of Following Pedestrian Violations Based on a Markov Chain

Many researchers focus on intersection crossing behavior using, e.g., the survey statistics method [15,16], the micro-simulation analysis method [17,18], the survival analysis method [19,20], the discrete choice method [21,22], and so on. Relevant studies have carried out in-depth analyses on the maximum time pedestrians will wait to cross the street, the delay caused by pedestrians crossing the street, and the traffic characteristics of speed, and have constructed models adapted to specific conditions. However, they fail to reflect the inherent following psychology and dynamic change process of pedestrians when crossing the street in a time series.

In this paper, the decision-making behavior of pedestrian crossing is divided into countable t states based on minutes. The time and state obtained are discrete, which conforms to the definition of a Markov chain. Therefore, the probability vector S(j) is constructed to represent the decision-making state of pedestrians in the *j*-th cycle:

$$S(j) = (S_1(j), S_2(j), S_3(j)) \tag{10}$$

where $S_t(j)$ (t = 1, 2, 3) represents the probability of being in state t during period j.

Other variables are defined as follows:

$n$ is the number of pedestrians counted by the survey sample; $A_i$ is the decision-making behavior sequence of the *i*-th pedestrian in the selected period, where $A_{i1}$ represents

the behavior sequence of pedestrians crossing the street in compliance, $A_{i2}$ represents the behavior sequence of pedestrians crossing the street in violation, $A_{i3}$ represents the behavior sequence of pedestrians following the previous violation, where $1 \leq I \leq n$.

In order to judge whether the decision-making sequence of pedestrian behavior conforms to the definition of a Markov chain, that is, the state of the random variable $X(n+1)$ only depends on the current state of $X(n)$, and is independent of the state of the previous random variable [23,24], the $\chi^2$ statistic can be used to test the "Markov property" of the discrete sequence [25]:

Suppose $v_1, v_2, \ldots v_n$ is a sequence of index values containing m states in the number of decision-making behaviors, using $f_{ij}$ to represent the frequency of a one-step transition from state $i$ to state $j$ in the sequence. The marginal probability $pj$ denotes the sum of the $j$-th column of the state transition frequency matrix F divided by the sum of each row and column of the matrix.

$$p_j = \frac{\sum_{i=1}^{m} f_{ij}}{\sum_{i=1,j=1}^{m} f_{ij}} \tag{11}$$

When $n$ tends to be large enough, the $\chi^2$ statistic is

$$\chi^2 = 2 \sum_{i=1,j=1}^{m} f_{ij} \left| \log \frac{p_{ij}}{p_j} \right| \tag{12}$$

which satisfies the $\chi^2$ distribution with freedom degrees $(m-1)^2$, where $p_{ij}$ is the transition probability from state $i$ to state $j$.

Under the given significance level $\alpha$, if the statistic $\chi^2$ satisfies

$$\chi^2 > \chi_\alpha^2 \left( (m-1)^2 \right)$$

it shows that the decision sequence of crossing behavior has a "Markov property". At this time, the k-step transition probability of Markov chain at time $n$ is

$$P\left\{ X(n+k) = j | X(n) = i \right\} = p_{ij}(n, k) \tag{13}$$

Suppose that the probability vector of a pedestrian's initial crossing state is $S(0) = (S_1(0), S_2(0), S_3(j))$ and, after k-step transition, the value is $S(k) = (S_1(k), S_2(k), S_3(k))$, that is, the selection behavior of pedestrian crossing after k cycles. In period $k$, the probability of pedestrian crossing in state $t$ is $S_t(k)(t = 1, 2, 3)$, and the transition probability matrix P of pedestrian crossing decision behavior state is

$$P(m) = \begin{bmatrix} p_{11} & p_{12} & p_{13} \\ p_{21} & p_{22} & p_{23} \\ p_{31} & p_{32} & p_{33} \end{bmatrix} \tag{14}$$

According to the ergodicity of the Markov chain, the probability of the process in state j is stable to $\pi$ after an infinite time from any state $\pi_j$. When $k \to \infty$, the state probability $S(k)$ will be infinitely close to a certain value, that is, as the transition step length gradually expands, the pedestrian's decision-making state will tend to be stable. In the steady state, the probabilities of the two decision states can be expressed by $\pi_1, \pi_2, \pi_3$:

$$\begin{cases} \pi = (\pi_1, \pi_2, \pi_3) \times P \\ \pi_1 + \pi_2 + \pi_3 = 1 \end{cases} \tag{15}$$

Since the statistical time series contains multiple statistical moments, the mean absolute error (MAE) and mean absolute percentage error (MAPE) can be used as evaluation indicators [26]. According to the relevant references, the smaller the values of these two

errors are, the more accurate the prediction results are, and the stronger the matching degree is:

$$\text{MAE} = \frac{\sum_{t=1}^{n} |\hat{qs} - qs|}{n} \tag{16}$$

$$\text{MAPE} = \frac{\frac{\sum_{t=1}^{n} |\hat{qs} - qs|}{qs}}{n} \tag{17}$$

where $\hat{qs}$ is the predicted value; qs is the actual value.

## 4. Case Analysis

### 4.1. Case Overview

This paper takes the Danxi Road-Lanxi Street intersection as an example, which is a cross signalized intersection, and the situation of each approach is shown in Figure 5.

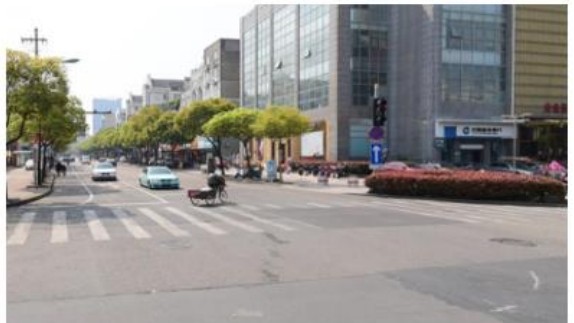 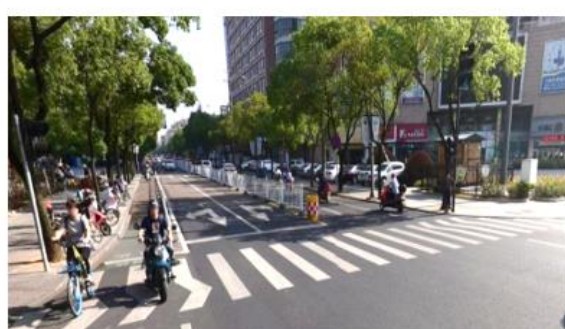

(**a**)

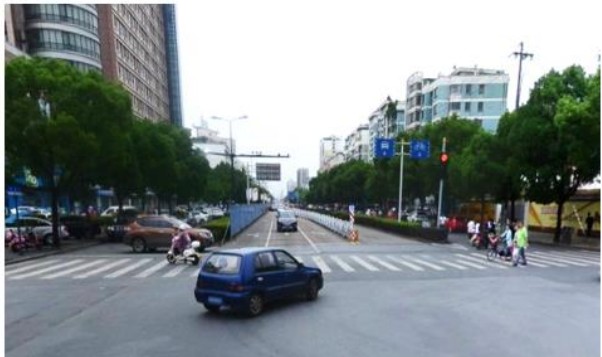 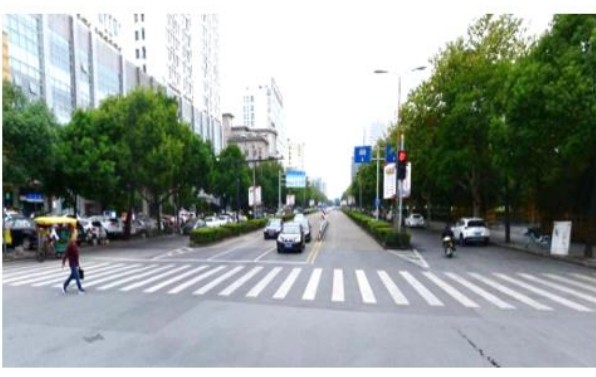

(**b**)

**Figure 5.** Video capture of the Danxi Road-Lanxi Street intersection. (**a**) Crosswalk going east–west. (**b**) Crosswalk going north–south.

Through the investigation, the basic traffic environment information of the Danxi Road-Lanxi Street intersection can be obtained and then transformed into corresponding index variables by the principal component analysis method. The survey data of 7:00–7:15 a.m. is shown in Table 8.

**Table 8.** Traffic environment information of the Danxi Road-Lanxi Street intersection.

| Second Level Indicators | Third Level Indicators | Index Value | Indicator Variables |
|---|---|---|---|
| F$_1$ | X$_1$ | 1 | 1.2 |
| | X$_2$ | 2 | |
| | X$_3$ | 1 | |
| F$_2$ | X$_4$ | 2 | 2 |
| | X$_5$ | 2 | |
| F$_3$ | X$_6$ | 1 | 1.5 |
| | X$_7$ | 2 | |

*4.2. Predictive Analysis and Feasibility Verification*

4.2.1. Data Collection

The pedestrian crossing survey was conducted from 7:00 to 8:00 a.m. in the morning rush hour, and the data in Table 9 were collected every 5 min.

**Table 9.** Number of pedestrians waiting to cross at the morning rush hour intersection.

| Time | Number of People Waiting to Cross | Number of Pedestrian Crossing Violations | Time | Number of People Waiting to Cross | Number of Pedestrian Crossing Violations |
|---|---|---|---|---|---|
| 7:00 | 28 | 9 | 7:30 | 29 | 9 |
| 7:05 | 39 | 11 | 7:35 | 45 | 15 |
| 7:10 | 26 | 8 | 7:40 | 35 | 11 |
| 7:15 | 37 | 12 | 7:45 | 42 | 14 |
| 7:20 | 35 | 11 | 7:50 | 33 | 12 |
| 7:25 | 32 | 10 | 7:55 | 29 | 9 |

The pedestrian crossing violations are divided into first pedestrian violations and following pedestrian violations, as shown in Table 10.

**Table 10.** Classified statistics of pedestrian crossing violations.

| Time | Number of First Pedestrian Violations | Number of Following Pedestrian Violations | Time | Number of First Pedestrian Violations | Number of Following Pedestrian Violations |
|---|---|---|---|---|---|
| 7:00 | 4 | 5 | 7:30 | 4 | 5 |
| 7:05 | 6 | 5 | 7:35 | 6 | 9 |
| 7:10 | 4 | 4 | 7:40 | 5 | 6 |
| 7:15 | 5 | 7 | 7:45 | 6 | 8 |
| 7:20 | 5 | 6 | 7:50 | 5 | 7 |
| 7:25 | 5 | 5 | 7:55 | 4 | 5 |

The statistics of crossing pedestrians at intersections under continuous time were mainly investigated according to the divided time intervals. In order to obtain the current state of pedestrians waiting to cross the street at the intersection, combined with the survey data in Table 9, Formula (18) obtained by the logistic model was used to process the survey data at 7:00–7:15 a.m. in Table 8.

$$p_1 = \frac{1}{1 + exp^{-0.296 - 0.534 \times 1.2 + 1.598 \times 2 - 0.429 \times 1.5}} = 0.165 \tag{18}$$

Therefore, the number of first pedestrian violations at the intersection could be predicted, as shown in Table 11.

### 4.2.2. Calculate the One-Step State Transition Probability Matrix

According to the survey statistics of pedestrian status, the frequency of different decision state transitions was obtained. Using the formula of the transition probability matrix [27] in discrete time, the one-step state transition probability matrix is calculated as follows:

$$P = \begin{bmatrix} 0.69 & 0.18 & 0.13 \\ 0.39 & 0 & 0.61 \\ 0.15 & 0.36 & 0.49 \end{bmatrix}$$

### 4.2.3. The Markov Property Test

The $\chi^2$ statistic can be used to test whether the discrete time series has "Markov properties". According to the one-step transfer probability matrix of pedestrians in discrete time obtained in the previous section, combined with Equation (7), under the given significance level $\alpha = 0.05$ when m = 4, we can obtain $\chi^2 = 15.316 > \chi^2_{0.05}\left((m-1)^2\right) = 12.854$, by querying the $\chi^2$ conditional distribution table. This result shows that the sequence has a "Markov property" and can be used to analyze and predict the future state.

### 4.2.4. Pedestrian Status Prediction

The one-step transition probability matrix was obtained by Step (2), and the pedestrian decision-making at multiple times of one continuous time period was predicted by the one-step transition probability $\pi$. Table 11 shows the predicted number of following pedestrian violations at the morning rush hour. Given the initial state (1,0,0), the prediction probability of pedestrian violations in the future can be obtained by iterative calculation. The comparison results with the actual violation probability are shown in Table 12:

**Table 11.** Forecast statistics of pedestrian crossing violations.

| Time | Number of First Pedestrian Violations | Number of Following Pedestrian Violations | Time | Number of First Pedestrian Violations | Number of Following Pedestrian Violations |
|------|------|------|------|------|------|
| 7:00 | 5 | 6 | 7:30 | 5 | 6 |
| 7:05 | 7 | 5 | 7:35 | 6 | 6 |
| 7:10 | 4 | 5 | 7:40 | 6 | 3 |
| 7:15 | 6 | 4 | 7:45 | 7 | 3 |
| 7:20 | 6 | 6 | 7:50 | 5 | 7 |
| 7:25 | 5 | 6 | 7:55 | 5 | 6 |

**Table 12.** Comparison of the actual and predicted number of pedestrian violations.

| Time | Actual Probability of Pedestrian Violations | Predicted Probability of Pedestrian Violations | Time | Actual Probability of Pedestrian Violations | Predicted Probability of Pedestrian Violations |
|------|------|------|------|------|------|
| 7:00 | 0.32 | 0.39 | 7:30 | 0.31 | 0.35 |
| 7:05 | 0.28 | 0.31 | 7:35 | 0.33 | 0.31 |
| 7:10 | 0.31 | 0.35 | 7:40 | 0.31 | 0.23 |
| 7:15 | 0.32 | 0.27 | 7:45 | 0.33 | 0.36 |
| 7:20 | 0.31 | 0.34 | 7:50 | 0.36 | 0.33 |
| 7:25 | 0.32 | 0.34 | 7:55 | 0.31 | 0.28 |

Based on the total number of pedestrians crossing at the intersection in Table 9, the actual survey data in Table 10 are compared with the prediction results in Table 11, so as to clarify the effect of the prediction model based on the Markov chain.

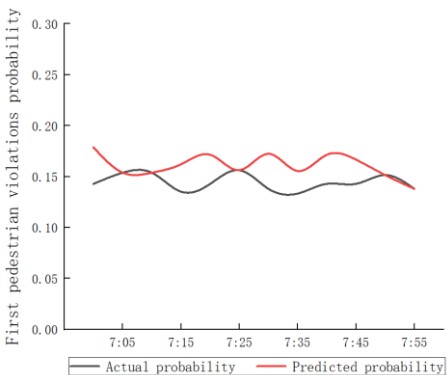

**Figure 6.** Comparison of the actual and predicted probability of first pedestrian violations.

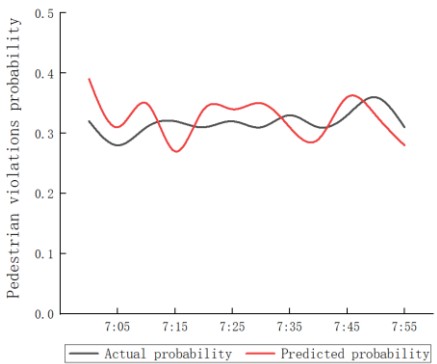

**Figure 7.** Comparison of the actual and predicted probability of following pedestrian violation.

Analyzing the Figures 6 and 7, it can be obtained that, considering that herd mentality has a large influence on whether a following pedestrian violation occurs, the predicted probability value curve has a larger variation range than the actual probability value curve when classifying the decision state of pedestrians. This method expands the prediction range and reflects the actual situation of pedestrians crossing the street. It shows that the Markov prediction model based on mean value can better predict the decision behavior of pedestrians crossing the intersection in peak hours. However, the use of Markov chain prediction has some limitations; that is, the number of pedestrians crossing in the actual process tends to be a random process, resulting in prediction that is limited to a certain time.

Since the Markov characteristics of pedestrians crossing the intersection determine the independence between two values, the predicted value may be higher than the actual value at certain moments. In addition, the prediction results show that the pedestrian crossing decision state has not changed greatly, which indicates that the pedestrian crossing decision state at the intersection is not optimal. Due to the large number of following pedestrian violations at the intersections, further optimization is needed to improve the pedestrian crossing compliance rate.

### 4.3. Prediction Results Evaluation

By evaluating the prediction effect of the established decision-making behavior model of pedestrians based on the mean Markov chain, it can be concluded that the average error of the number of waiting pedestrians at the intersection is 3.18 person-times, and the average relative error percentage is only 0.97745%, less than 1%. This indicates that the prediction ability of the model is good under the given conditions, so the model can effectively predict pedestrian decision-making at intersections at certain continuous times.

## 5. Conclusions

This paper takes pedestrian crossing at urban signalized intersections as the research object. Combined with field investigation and video recording detection methods, the quantifiable indicators affecting pedestrian crossing behavior are preliminarily classified.

According to the pedestrian crossing behavior data collected from the survey, the index system affecting the decision-making behavior acts by the first pedestrian is constructed, and parameters are calibrated by the logistic regression analysis model. The evaluation results show that the model can well reflect the behavior characteristics of the first pedestrian crossing at the signalized intersection.

Based on the analysis of the mechanism of the first pedestrian's crossing violation, the decision model of the following pedestrian crossing is constructed by combining herd psychology and the Markov chain method. The results can predict the probability of pedestrian crossing violations at intersections. Finally, the validity of the logistic regression analysis model and the Markov chain method is verified by the survey data of pedestrians crossing at the Danxi Road-Lanxi Street intersection in Jinhua City.

Generally, some limitations in this study merit further study. Firstly, the psychological characteristics that affect pedestrian crossing violations should be further explored to refine the influencing factors; secondly, the model assumes that the indicator variables can be obtained by actual observation, a limitation that can be handled by a Hidden Markov Model; thirdly, the research results can be extended for a traffic flow assessment [28,29] to improve road safety analysis [30,31]. Finally, more pedestrian crossing data from multiple types of intersections are needed to validate the proposed model in this paper.

**Author Contributions:** C.M. designed the study; L.B. conducted data analysis and wrote the report; S.Y. and W.X. collected the data; Q.W. contributed to the rationale and discussion of results. All authors have read and agreed to the published version of the manuscript.

**Funding:** This research was funded by the Natural Science of Zhejiang Province LY18G030021.

**Institutional Review Board Statement:** Not applicable.

**Informed Consent Statement:** Not applicable.

**Data Availability Statement:** Data cannot be shared publicly because of the confidentiality of the Traffic Police Brigade of Jinhua, Zhejiang, China, who imposed data sharing restrictions on the data underlying our study. Data are available for researchers who meet the criteria for access to confidential data.

**Conflicts of Interest:** The authors declare no conflict of interest.

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
