# Peer review of "Analysis and Prediction of Pedestrians’ Violation Behavior at the Intersection Based on a Markov Chain"

_sustainability, doi:10.3390/su13105690_

Round 1
Reviewer 1 Report
This paper addresses an important and interesting problem - pedestrians crossing intersections illegally. Based on herd psychology, this paper studies the characteristics of pedestrians crossing the street, using the Logistic model and Markov chain method to predict the violation probability of the first pedestrian and following pedestrians respectively. The results are obviously helpful in improving the safety performance of intersections.
Overall, the article is well organized and interesting. However, some minor issues still need to be improved:
- In section 1, I suggest introducing the differences between the first pedestrian and the following pedestrians;
- I suggest that the accuracy and reliability of the proposed model are compared with other pedestrian violation probability prediction models;
- In Figure 7. on Pg. 12, I think the curve reflects the violation characteristics of the following pedestrians, not all pedestrians;
- There are a few typos and grammar errors in this paper.
- The limitations of this study and the future research directions are suggested to provide. For instance, the pedestrian violation data can be collected with more advanced techniques, such as computer vision techniques.
Reviewer 2 Report
The article is well written. The authors are suggested to add more literature review and provide the shortcoming in your article and provide suggestion to improve it in the conclusion section.
Reviewer 3 Report
The authors mainly proposed a Markov chain method to predict the occurrence of pedestrian violations at signalized intersections. It is an interesting paper. With the field data collection, affecting factors of crossing behaviors were analyzed, and the model test and case study were also completed. However, some content of the paper should be revised and improved before accepting this paper. Detailed comments are below:
- Literature review of related studies is not enough stated in the introduction section.
- Line 88, I didn't see the value of this statement.
- In Figure 4, keep a consistent format for the subscript numbers.
- Line 178, please rewrite this paragraph.
- Line 188, is it the significance of the three indictors (F1, F2 and F3) or of the variables within each indicator? Please verify it. It is also not clear that how the authors use X1, X2 etc to represent F1, F2 and F3 in the logistic model.
- For Eq. (5), what do V, R, and A mean? Isn't it beta_3 instead of beta_2 associated to A?
- Line 239, it is not clear about the definition of state t in the model.
- Please check carefully the figure captions through the paper, such as Figure 2, ... violation in time; Figure 3.
- Some typos in Eqs. (15) and (16): MAE instead of MEA; MAPE instead of MEPA; and the proper word of "where" rather than "Where:" follows the equations.
- Is Table 10 the same as Table 9?
Round 2
Reviewer 2 Report
Maybe accepted